# Older adults select different but not simpler strategies than younger adults in risky choice

**Florian Bolenz**[1,2], **Thorsten Pachur**[1,2,3]

1 Center for Adaptive Rationality, Max Planck Institute for Human Development, Berlin, Germany, 2 Science of Intelligence, Research Cluster of Excellence, Berlin, Germany, 3 School of Management, Technical University of Munich, Munich, Germany

* bolenz@mpib-berlin.mpg.de

## Abstract

Younger and older adults often differ in their risky choices. Theoretical frameworks on human aging point to various cognitive and motivational factors that might underlie these differences. Using a novel computational model based on the framework of resource rationality, we find that the two age groups rely on different strategies. Importantly, older adults did not use simpler strategies than younger adults, they did not select among fewer strategies, they did not make more errors, and they did not put more weight on cognitive costs. Instead, older adults selected strategies that had different risk propensities than those selected by younger adults. Our modeling approach suggests that age differences in risky choice are not necessarily a consequence of cognitive decline; instead, they may reflect motivational differences between age groups.

## Author summary

What are the psychological mechanisms underlying adult age differences in economic decision making? We investigated this question with a model based on the framework of resource rationality, which posits that people adaptively use the cognitive resources available to them. Unlike commonly used economic models of decision making, this model can shed light on the cognitive processes that drive age differences in choice. Our findings show that younger and older adults use different decision strategies and that the age differences are not necessarily the result of cognitive decline; instead, they may be a result of age differences in motivational factors. By providing novel insights into the psychological mechanisms of age differences in decision making, our modeling approach can inform interventions and choice architectures supporting older adults' decision making.

## Introduction

Decision making differs between younger and older adults. In decisions under risk, older adults have often been found to show lower decision quality than younger adults—that is, to be less likely to choose the option with the higher expected value [1]. Age differences in risk

**Data Availability Statement:** Data and scripts are publicly available at https://osf.io/4sqcj/.

**Funding:** Funded by the Deutsche Forschungsgemeinschaft (DFG, German Research Foundation) under Germany's Excellence Strategy

– EXC 2002/1 "Science of Intelligence" – project number 390523135 (TP). The funders had no role in study design, data collection and analysis, decision to publish, or preparation of the manuscript.

**Competing interests:** The authors have declared that no competing interests exist.

aversion have also been observed—that is, in how often a decision maker chooses the option with the less variable range of possible outcomes. Findings on the direction of age differences in risk aversion are heterogeneous, depending also on the characteristics of the decision task [2]. For instance, whereas in choices between a risky and a safe option older adults are more risk averse than younger adults [3, 4], in choices between two risky options they are equally or even more risk seeking [5–9].

What drives age differences in risky decision making? Several lines of research point to cognitive and motivational changes across the lifespan that might also be relevant for decision making and thus help explain the observed age differences in risky choice. In terms of age-related cognitive decline, for instance, it has been concluded that older adults rely on simpler cognitive strategies (e.g., in reinforcement-learning, judgment, or working-memory tasks [10–14]) and make more errors [1] than younger adults. Moreover, older adults show less flexibility in adjusting cognitive processes to the situation [13, 15–17] and they may make different trade-offs between the complexity of a cognitive process and its potential benefits [18]. These changes due to age-related cognitive decline might affect the mental operations selected by a decision maker when making risky choices.

Age-related changes in motivational factors involved in decision making may also help explain age differences in risky choice. For instance, older adults report experiencing more positive and less negative affect than younger adults [19, 20], and precisely this pattern of more positive and less negative affect has been linked to increased risk taking [21, 22]. Further, older adults attend more strongly to positive than to negative information [23] and they focus more on positive emotions when making risky choices than do younger adults [3, 24]. Finally, older adults differ from younger adults in their sensitivity to potential gains and losses [25–27] and it has been proposed that the motivation to prevent losses increases with age [28].

To date, a rigorous application of these theoretical perspectives on aging to decisions under risk has been difficult. Age differences in risky choice are commonly modeled with expected utility theory and extensions thereof, such as cumulative prospect theory [29] (see, e.g., [1, 2, 4, 5, 7]). It is challenging, however, to link age differences measured with expected utility models to age-related changes in cognition, affect, and motivation. The reason is that expected utility theories are premised on psychoeconomic curves that describe how the mapping of objective outcomes and probabilities onto subjective values deviates from what is mandated by normative accounts. However, these psychoeconomic curves are not intended to describe psychological processes [30, 31]. Consequently, it is not exactly clear how age-related cognitive and motivational differences would be reflected in models in the expected utility tradition, which makes it difficult to test the different theoretical perspectives on psychological factors potentially underlying age-related differences in risky choice.

Recently, the theory of resource-rational strategy selection [32] has been proposed to describe the adaptive use of different cognitive strategies in decision making. Building on ideas of Payne, Bettman & Johnson [33], this theoretical framework assumes that decision makers are equipped with a toolbox of cognitive strategies from which they select a strategy for a given choice problem based on cost–benefit considerations. These strategies can follow different simplifying principles—for example, limiting evaluation to given aspects of the choice options or specifying how information is integrated during the evaluation of choice options [34, 35]. They thus differ in their cognitive costs and ability to identify the option with the higher expected payoff. According to the theoretical framework of resource-rational strategy selection, the decision maker selects that strategy that in a given choice problem strikes the best balance between the expected payoff and the cognitive cost of implementing the strategy—in other words, the decision maker makes optimal use of their finite cognitive resources. Individuals can differ in their cognitive resources and therefore put different weights on the cognitive

costs during strategy selection. For example, when cognitive resources are experimentally constrained, people shift to simpler decision strategies [36, 37], which can produce differences in risk aversion [37]. In contrast to models in the expected utility tradition, models of cognitive strategies can be interpreted as accounts of cognitive processes [33, 35, 38]. The framework of resource-rational strategy selection thus allows for a more seamless connection between psychological theories of aging and models of decision making under risk by helping to disentangle the cognitive and motivational factors that impact strategy selection.

In this article, we apply, to our knowledge for the first time, the framework of resource-rational strategy selection to model age differences in risky choice. We re-analyze data by Pachur, Mata & Hertwig [5], in which 60 younger and 62 older adults made risky choices in a set of 105 choice problems (mostly consisting of two risky options). In additional tasks, older adults showed lower fluid cognitive abilities and reported less negative and more positive affect than younger adults. We examined five hypotheses for the risky choice data (all but the first and last were preregistered at osf.io/k9sx2). The first hypothesis assumes that age differences in risky choice reflect differences in strategy use, without distinguishing between cognitive and motivational factors driving these differences. The other four hypotheses follow from the notion of age-related cognitive decline.

*Strategy-distribution hypothesis*. There are qualitative differences in the distribution of strategies selected by younger and older adults.

*Strategy-complexity hypothesis*. Older adults use less complex strategies than younger adults [10–12, 14].

*Toolbox-size hypothesis*. Older adults have fewer strategies in their mental toolboxes than younger adults. Because the computational demands of strategy selection may increase with the number of cognitive strategies available [39], limiting the number of strategies should reduce cognitive demands.

*Strategy-selection hypothesis*. Due to their lower cognitive resources [40], older adults put more weight on the cognitive cost of a strategy during strategy selection than do younger adults.

*Strategy-execution hypothesis*. Older adults are more error-prone in executing strategies [41–43].

## Results

In a first step, we characterized participants' choices in terms of decision quality and risk aversion. To quantify decision quality, we determined for each choice problem whether a participant had chosen the option with the higher expected value (which reflects the long-term payoff of choosing the respective option). To quantify risk aversion, we determined for every trial whether a participant had chosen the option with the lower coefficient of variation [44]. We used mixed-effects logistic regression to examine whether younger and older adults differed in decision quality and risk aversion, with either decision quality or risk aversion as the dependent variable and with age group (younger vs. older), problem domain (gain, loss, or mixed) and their interaction as fixed effects; we included random intercepts for participants and choice problems. Reproducing the results by Pachur, Mata & Hertwig [5], the analyses showed that older adults had lower decision quality than younger adults in loss problems ($b = -0.33$, 95%credible interval (CI) = $[-0.54, -0.11]$, Cohen's $d = -0.19$; see bars in Fig 1), and lower risk aversion than younger adults in gain problems ($b = -0.42$, CI = $[-0.60, -0.23]$, $d = -0.23$) and mixed problems ($b = -0.33$, CI = $[-0.52, -0.14]$, $d = -0.19$, see bars in Fig 1).

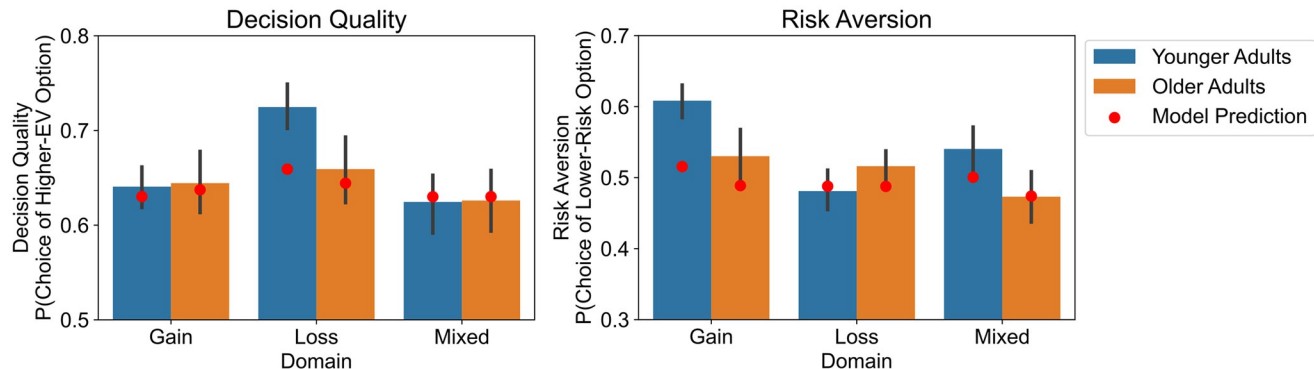

**Fig 1. Decision quality (left) and risk aversion (right) by problem domain and age group.** Bars show the empirically observed behavior (error bars represent the standard error of the mean), red dots show the average predictions of the model simulations.

To examine how these differences between younger and older adults in risky choice are reflected in the resource-rational strategy selection model, we applied the model to each individual's choice data. According to the model, choices result from the use of different cognitive strategies that are selected from a toolbox of strategies contingent on the properties of the current choice problem. We considered 11 strategies that have been proposed for risky choice (Table 1); the strategies differ in complexity and have been shown to give rise to systematically different degrees of risk aversion [45]. A key assumption of the resource-rational strategy selection model is that the decision maker selects a strategy for each choice problem by weighting its expected payoff against its cost. The weighting of the strategy's cost is controlled by the cost-weighting parameter $\delta$; larger values of $\delta$ reflect a larger weight being given to strategy cost during strategy selection. The selected strategy is then executed with a trembling-hand error (expressing the proportion of cases in which an option other than that predicted by the strategy is chosen), to account for potential noise in the application of the strategy [46]. All model parameters (including the size and the composition of the strategy toolbox) were estimated from the data for each participant and are assumed to be constant across choice problems.

To examine to what extent the resource-rational strategy selection model was able to capture the participants' choices, we conducted posterior predictive checks. For this purpose, we used each individual's best-fitting parameter values and toolboxes to simulate choices in the choice task; as the model predicts a choice probability (that is governed by the trembling-hand error parameter), we repeated this 100 times. For each participant, we computed the proportion of trials in which the choice predicted by the model matched the empirically observed choice. The proportion of matching choices was, on average, 0.62 in both younger adults (range: 0.52–0.80) and older adults (range: 0.51–0.89). A two-sided Bayesian $t$-test indicated moderate evidence that the match of the simulated with the actual choices did not differ between age groups ($BF_{10} = 0.22$). In other words, the resource-rational strategy selection model captured the choices of both age groups equally well.

As a further test of model fit, we assessed how well the resource-rational strategy selection model captured the empirically observed pattern of age differences in decision quality and risk aversion. To that end, we analyzed the choices simulated by the model based on the best-fitting parameter values in terms of decision quality and risk aversion (averaged across the 100 model simulations) with a mixed-effects beta regression with age group (younger vs. older), problem domain (gain, loss, and mixed), and their interaction as fixed effects, and random intercepts

Table 1. Proposed Cognitive Strategies for Risky Choice [34, 35].

| Strategy | Description |
|---|---|
| Minimax | Choose the option with the highest minimum outcome. |
| Maximax | Choose the option with the highest outcome. |
| Least-likely | Identify each option's worst outcome. Then choose the option with the lowest probability of the worst outcome. |
| Most-likely | Identify each option's most likely outcome. Then choose the option with the highest most likely outcome. |
| Better-than-average | Calculate the grand average of all outcomes from all gambles. For each gamble, count the number of outcomes equal to or above the grand average. Then select the gamble with the highest number of such outcomes. |
| Equal-weight | Calculate the sum of all outcomes within a gamble. Choose the gamble with the highest sum. |
| Tallying | For gamble problems in the gain domain, give a tally mark to the gamble with (a) the higher minimum gain, (b) the higher maximum gain, (c) the lower probability of the minimum gain, and (d) the higher probability of the maximum gain. For gamble problems in the loss domain, replace "gain" by "loss" and "higher" by "lower" (and vice versa). Select the gamble with the highest number of tally marks. |
| Probable | Categorize probabilities as "probable" (i.e., $p \geq .5$ for a two-outcome gamble) or "improbable." Cancel improbable outcomes. Then calculate the arithmetic mean of the probable outcomes for each gamble. Finally, select the gamble with the highest average payoff. |
| Lexicographic | Determine the most likely outcome of each gamble and their respective payoffs. Then select the gamble with the highest most likely payoff. If all payoffs are equal, determine the second most likely outcome of each gamble and select the gamble with the highest (second most likely) payoff. |
| Priority heuristic | Go through attributes in the following order: minimum gain, probability of minimum gain, and maximum gain. Stop examination if the minimum gains differ by 1/10 or more of the maximum gain; otherwise, stop examination if the probabilities differ by 1/10 or more of the probability scale. Choose the option with the most attractive gain (probability). |
| Weighted-additive | For each gamble, sum up the possible outcomes weighted by their probabilities. Choose the option with the highest weighted sum. |

*Note.* Descriptions of strategies are adopted from [47].

for participants and choice problems (similar to the structure of the regression models used to analyze the empirical data). Mirroring the empirical findings, the simulated choices of older adults showed lower decision quality than those of younger adults in loss problems, although the 95% credible interval did not exclude 0 ($b = -0.05$, CI = [−0.19, 0.09], $d = -0.05$; see dots in Fig 1). Furthermore, the simulated choices of older adults showed lower risk aversion than those of younger adults in gain problems ($b = -0.12$, CI = [−0.20, −0.03], $d = -0.10$) and mixed problems ($b = -0.12$, CI = [−0.21, −0.03], $d = -0.10$). In sum, the resource-rational strategy selection model captured the age differences in risky choice, although the differences as reflected in the model were somewhat less pronounced than the empirically observed ones.

Additionally, we compared for each participant their empirically observed decision quality and risk aversion (pooled across all three problem domains) against the decision quality and risk aversion of the choices simulated with their best-fitting model parameters (Fig 2). The higher a participant's decision quality, the higher was the decision quality of their simulated choices ($r = 0.91$, $BF_{10} = 5.7 \times 10^{41}$). Moreover, the higher a participant's risk aversion, the higher was their simulated risk aversion ($r = 0.79$, $BF_{10} = 3.1 \times 10^{23}$). This analysis shows that the resource-rational strategy selection model captured the individual differences between participants in choice behavior well.

We also compared the resource-rational strategy selection model with cumulative prospect theory [29], arguably the most prominent model for risky choice. Cumulative prospect theory

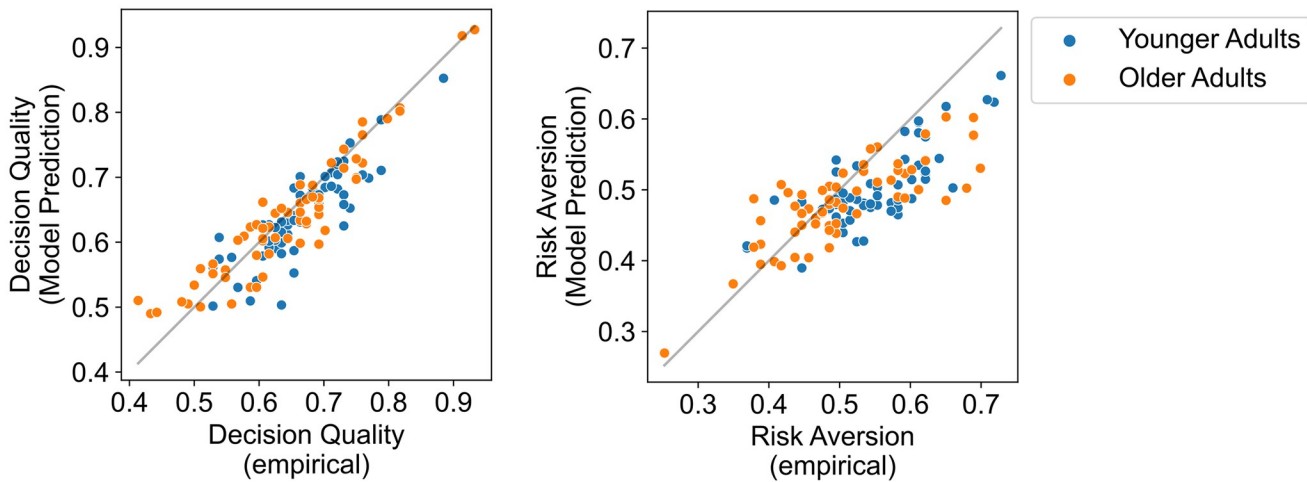

**Fig 2. Comparison of simulated (based on the fitted resource-rational strategy selection model) and empirically observed levels of decision quality (left) and risk aversion (right).** Each circle represents one participant (with the choices pooled across the three domains; i.e., gain, loss, and mixed). The diagonal lines indicate identity (i.e., perfect fit).

showed a better fit for the choice data in terms of higher log-likelihoods (Table 2). While this pattern was evident in both age groups, it was more pronounced for the younger adults than for the older adults. This analysis does not take into account differences in model complexity because quantifying the number of free parameters of the resource-rational strategy selection model is not straightforward due to the categorical, high-dimensional nature of the strategy toolbox parameter. With that limitation in mind, we can conclude that cumulative prospect theory describes the choice data better than the resource-rational strategy selection model. Importantly, however, cumulative prospect theory does not allow insights into the cognitive processes underlying people's choices.

## Do younger and older adults rely on different strategies?

To assess possible age differences in strategy use, we used the best-fitting parameters of the resource-rational strategy selection model for each individual to compute how often the selection of each strategy was predicted in the two age groups (Fig 3). We normalized the strategy counts by dividing each count by the number of simulation runs. A Bayesian contingency-table test showed that there was strong evidence that the distribution of strategies differed between age groups ($BF_{10} = 1.6 \times 10^{163}$, Cramér's $V = .25$). According to the estimated model, the most frequently used strategies were the minimax heuristic, the least-likely heuristic, the priority heuristic, the equal-weight heuristic, and the maximax heuristic. Younger adults were estimated to rely more frequently than older adults on the minimax heuristic, the least-likely heuristic, and the priority heuristic; older adults were estimated to rely more frequently than younger adults on the equal-weight heuristic and the maximax heuristic. Note that the

**Table 2. Log-likelihoods of the resource-rational strategy selection model and cumulative prospect theory.** Higher values indicate a better fit.

| Model | Full sample | Younger adults | Older adults |
|---|---|---|---|
| Resource-rational strategy selection | -7236 | -3544 | -3692 |
| Cumulative prospect theory | -6666 | -3025 | -3640 |

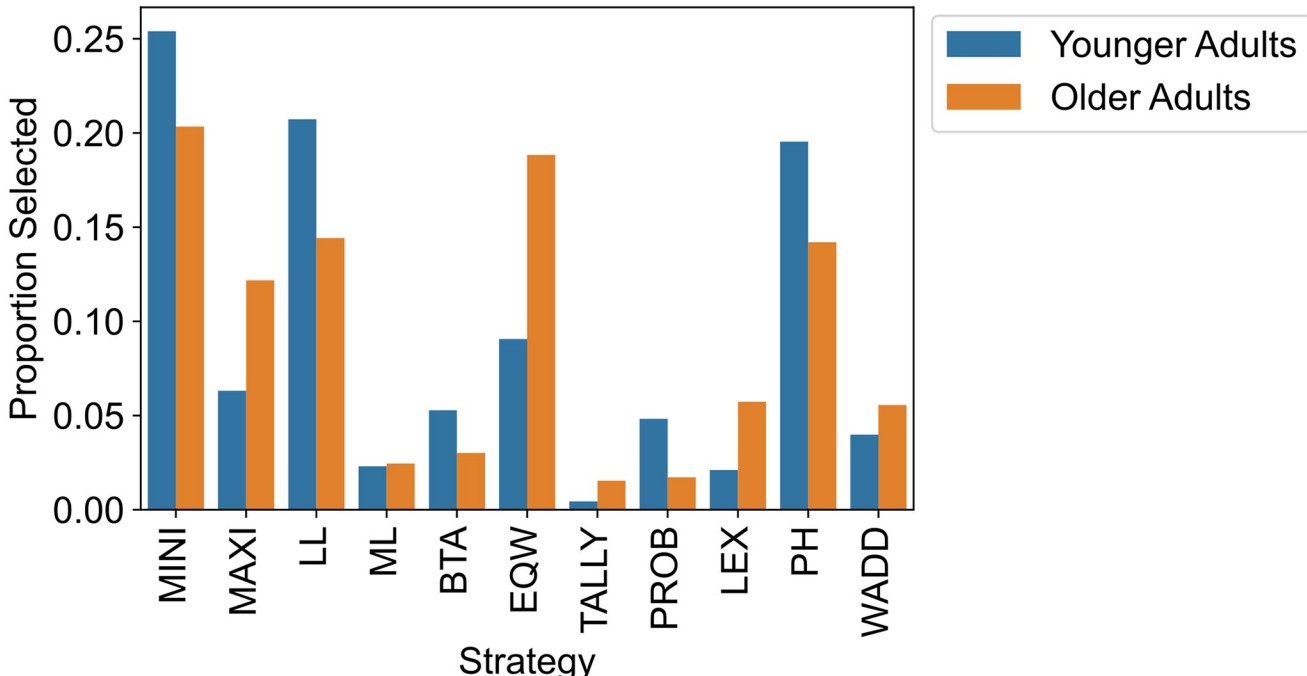

**Fig 3. Frequency of strategies across all trials.** MINI: Minimax, MAXI: Maximax, LL: Least-likely, ML: Most-likely, BTA: Better-than-average, EQW: Equal-weight, TALLY: Tallying, PROB: Probable, LEX: Lexicographic, PH: Priority heuristic, WADD: Weighted-additive.

strategies more frequently selected by older adults only consider information about outcomes and not about probabilities, whereas the strategies more frequently selected by younger adults tend to consider both outcome and probability information; this could point to a more general difference in information search between the age groups. These results suggest that—consistent with our strategy-distribution hypothesis—older adults used a somewhat different set of strategies in their risky choices than younger adults.

To examine to what extent these different strategy sets can give rise to the empirically observed age differences in risk aversion [45], we analyzed the risk profiles of the most frequently selected strategies (maximax, equal-weight, minimax, least-likely, priority heuristic; see S1 Text for details). Specifically, we determined the tendency of these strategies to choose the less risky option (i.e., the one with the lower coefficient of variation) in each choice problem. Indeed, the risk profiles of these strategies echoed the pattern of age differences in risk aversion: The strategies more frequently selected by older adults (i.e., maximax, equal-weight) showed lower risk aversion in gain and mixed problems than the strategies more frequently selected by younger adults (i.e., minimax, least-likely, priority heuristic). These findings suggest that the observed age differences in risk aversion may indeed be directly attributable to differences in the selection of specific strategies.

## Do older adults rely on simpler strategies than younger adults?

Next, we investigated whether the differences in strategy use might be due to older adults using simpler strategies than younger adults. To that end, we compared the average cost of the strategies that were estimated by the model to be selected by older and younger adults. As in previous applications of the resource-rational strategy selection model [32, 39, 48], we

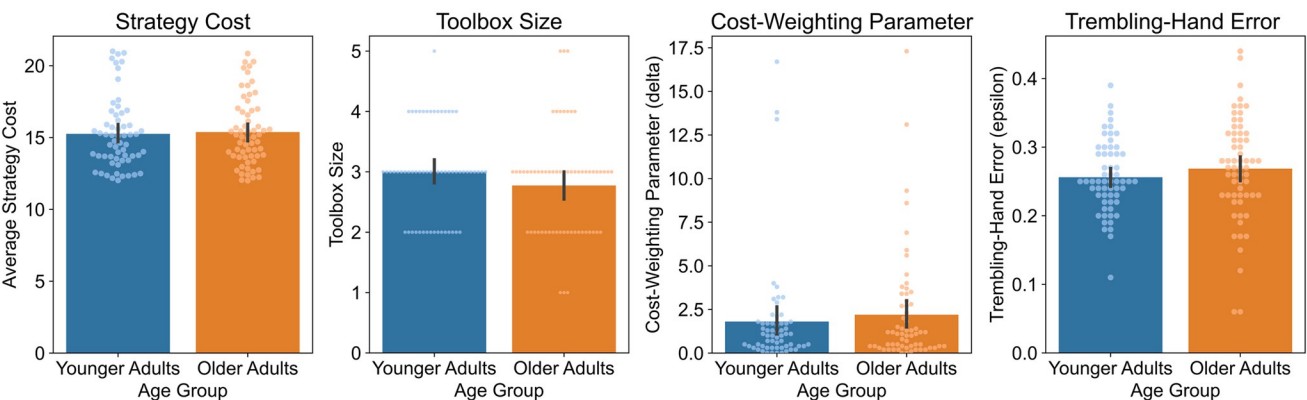

**Fig 4. Average strategy cost (A), toolbox size (B), cost-weighting parameter (C) and trembling-hand error (D) for younger and older adults as estimated with the resource-rational strategy selection model.** Bars indicate group means, with error bars representing the standard error of the mean. Points represent individual participants.

operationalized strategy cost as the number of mental operations required by a strategy [49]. For younger adults, the average strategy cost was 15.26 ($SD$ = 2.46); for older adults, it was 15.38 ($SD$ = 2.36) (Fig 4A). A one-sided Bayesian $t$-test showed moderate evidence that the age groups did not differ in terms of the costs incurred by the strategies used ($BF_{10}$ = 0.16). In other words, contrary to our strategy-complexity hypothesis, older adults did not seem to rely on simpler strategies than younger adults.

### Do older adults rely on smaller toolboxes than younger adults?

Our next test examined whether older adults have a smaller toolbox of strategies at their disposal than younger adults. According to the estimated resource-rational strategy selection model, younger adults' toolboxes contained, on average, 3.00 ($SD$ = 0.76) strategies and older adults' 2.77 ($SD$ = 0.89) strategies (Fig 4B). A one-sided Bayesian $t$-test indicated inconclusive evidence that the older adults have fewer strategies in their mental toolbox than younger adults ($BF_{10}$ = 0.98). That is, contrary to the toolbox-size hypothesis, there was no evidence that older adults selected from a smaller set of strategies than did younger adults.

### Do younger and older adults differ in how they trade off costs and accuracy during strategy selection?

To investigate whether younger and older adults differ in how they trade off payoff and costs in strategy selection, we compared the cost-weighting parameter $\delta$ (see Eq 1) estimated for the two age groups. The parameter $\delta$ reflects the weight of strategy cost against the expected payoff of a strategy. The estimated parameter was, on average, $\delta$ = 1.81 ($SD$ = 3.14) for younger adults and $\delta$ = 2.20 ($SD$ = 3.17) for older adults (Fig 4C). A one-sided Bayesian $t$-test indicated inconclusive evidence for the hypothesis that the cost-weighting parameter is larger for older than for younger adults ($BF_{10}$ = 0.36). That is, contrary to the strategy-selection hypothesis, there was no evidence that older adults put more weight on the cognitive cost of a strategy during strategy selection than do younger adults.

### Are older adults more error-prone in strategy execution?

Finally, we tested whether there was more noise in older adults' than younger adults' execution of the strategies; this would be reflected in higher values of the trembling-hand error parameter

$\epsilon$. The average trembling-hand error was $\epsilon = 0.26$ ($SD = 0.05$) for younger adults and $\epsilon = 0.27$ ($SD = 0.08$) for older adults (Fig 4D). A one-sided Bayesian $t$-test indicated inconclusive evidence for the hypothesis that the trembling-hand error is higher for older adults ($BF_{10} = 0.55$). Thus, contrary to the strategy-execution hypothesis, older adults did not seem to be more error-prone in their execution of the strategy selected than younger adults.

## Discussion

We applied a computational model of resource-rational strategy selection [32] to investigate psychological factors that drive age differences in risky choice. In particular, we were interested in whether older adults simplify the decision making process—potentially due to cognitive decline. Our results suggest that older adults used qualitatively different strategies than younger adults; however, there was no evidence that these differences were a consequence of age-related cognitive decline: Older adults did not select among fewer different strategies, they did not use less complex strategies than younger adults, they did not put more weight on strategy cost, and they did not commit more errors during strategy execution. Instead, the age differences in decision making resulted from different configurations of participants' strategy toolboxes.

If the age differences in strategy use are not due to cognitive factors, what else might drive them? One possibility is that they reflect motivational differences. Older adults typically report more positive affect than younger adults, as observed in the analyzed dataset [5], and more positive affect has been associated with lower risk aversion [22]. A mediation analysis showed that around 30% of the age effect on risk aversion captured by the differences in strategy selection can be attributed to age differences in positive and negative affect—consistent with a motivational account of the age differences in strategy selection (see S2 Text).

While our analysis suggests that cognitive factors do not play a substantial role in age differences in risky choice, previous research has concluded that cognitive decline accounts for age differences in other domains of decision making. In reinforcement-learning tasks, for example, it has been suggested that older adults use simpler strategies than younger adults because they have difficulties in learning and maintaining an accurate representation of the latent task structure, which is an important prerequisite for the use of more complex strategies [50]. One possible explanation for cognitive factors not playing a substantial role in the present data is that, in contrast to reinforcement-learning tasks, all relevant information was directly observable in the risky choice tasks; arguably, this facilitates the implementation even of complex strategies. In line with this possibility, age differences in risky choice seem to be more pronounced when the decision task involves working memory [51].

The computational framework of resource-rational strategy selection offers a new perspective on age differences in risky choice. In contrast to approaches commonly used to model age differences in risky choice, such as expected utility theory [1] or cumulative prospect theory [5, 7], findings from the resource-rational strategy selection model are directly interpretable in terms of cognitive information processing. Therefore, the resource-rational strategy selection model is able to provide novel insights into the psychological processes underlying age differences in risky choice even though in its current formalization it does not capture the choice behavior as well as models based on expected utility theory. As demonstrated with our current analyses, the resource-rational strategy selection model allows one to derive and test various hypotheses on different ways in which cognitive decline might affect cognitive processing. Further, being based on models of cognitive strategies, the resource-rational strategy selection model led to the observation that compared to younger adults, older adults seem to be less likely to process both outcome and probability information. An intriguing issue for future

research is to examine the extent to which age-related differences in strategy selection as identified here are related to age differences in patterns of predecisional information search (e.g., eye tracking, information boards; [52]). Further, insights into the psychological processes that underlie decision making in different age groups can inform age-specific interventions that help people make good decisions. For example, following the observation that older adults were more likely than younger adults to select strategies that considered information about outcomes but not their probabilities, interventions increasing the attention paid to probability information [53] could prompt older adults to make greater use of strategies that consider probability information—and thus help them make better choices.

The resource-rational strategy selection model is able to capture differences in the direction of the effect of age between gain, loss, and mixed problems without assuming parameter differences between domains. This suggests that the apparently complex interaction between age group and domain on risk aversion in the empirical data may be due to simple differences in strategy use—which in turn produce different risk propensities even if strategy selection is invariant across domains. Still, allowing strategy selection to differ between domains might even further improve the fit of the model to the empirical data (e.g., aligning the model predictions more closely with the empirically observed behavior of younger adults with respect to decision quality in the loss domain and risk aversion in the gain domain). There is evidence that people invest more cognitive resources in problems involving losses [54]; they might therefore make different trade-offs between payoff and cost as a function of the problem domain. Estimating the model parameters reliably for each of the choice domains separately would, however, require more data per participant than are available in the dataset analyzed in the current study.

Our findings should be interpreted in the light of the following limitations. First, although we considered a comprehensive set of decision strategies [34, 35], we do not claim that it is exhaustive. An alternative approach would be to identify decision strategies in a data-driven way from process data [55] (but see [47] for a more critical perspective). Second, our operationalization of strategy cost follows the framework of elementary information processes [49], as is common in implementations of the resource-rational strategy selection model [32, 39, 48]. While this approach provides a useful proxy for strategy cost, more complex aspects of strategy cost are conceivable (e.g., differences in the costs of processing probabilities vs. outcomes; [56]), and quantifying cost more precisely remains an important task for future research [57–59]. Improving the set of decision strategies and the operationalization of strategy cost could further increase the fit of the model.

In conclusion, using a computational framework that allows us to test the possible consequences of age-related cognitive decline on strategy selection, we found that older adults use different but similarly complex decision strategies as younger adults in risky choice. Modeling risky choices in terms of cognitive strategies offers a promising approach with insights that are not available from currently dominant modeling accounts of risky choice and can contribute substantially to the understanding of the psychological underpinnings of age differences in decision making.

## Method

### Dataset

We re-analyzed data from Pachur, Mata & Hertwig [5]. This dataset contains choice data from 60 younger adults (46 female, 14 male, mean age: 23.6 years, range: 18–30 years) and 62 older adults (31 female, 29 male, 2 who did not report their gender, mean age: 71.3 years, range: 63–88 years). Participants completed 105 risky choice problems. Of the problems, 41 were in the

gain domain, 31 were in the loss domain, and 33 were mixed. Ninety-six of the problems consisted of two risky gambles, each with two outcomes and corresponding probabilities, 9 problems also involved a safe option.

We also report the analysis of a second dataset on age differences in risky choice, collected by Horn, Schaltegger, Best & Freund [7] (see S3 Text); this analysis replicates the conclusion that age differences in strategy selection between younger and older adults are not due to cognitive factors.

## Resource-rational strategy selection model

The resource-rational strategy selection model [32] assumes that a decision maker is equipped with a set $S$ of potential strategies. For every choice problem $p$, the decision maker selects the strategy $s^*$ that optimizes the trade-off between the expected strategy payoff $r_{s,p}$ and the expected strategy cost $c_{s,p}$.

$$s^* = \underset{s \in S}{\operatorname{argmax}} \left( r_{s,p} - \delta \cdot c_{s,p} \right) \tag{1}$$

The weighting parameter $\delta$ governs how much weight is given to the cost term $c_{s,p}$. With $\delta = 0$, the amount of cognitive resources available plays no role in strategy selection and the decision maker simply selects the strategy with the highest expected payoff. With higher values of $\delta$, the cost of a strategy has a stronger impact on strategy selection.

In our analyses, we consider a comprehensive set of cognitive strategies previously suggested for risky choice ([34, 35]; see Table 1 for a detailed description). Most of these strategies have clearly different decision profiles for the choice problems in the analyzed dataset [5] (see S4 Text). The resource-rational strategy selection model considers these strategies as candidate elements in each participant's toolbox. To quantify the expected payoff and cost of a strategy $s$ in a given choice problem $p$, we simulated the choices of each strategy for each of the choice problems. The expected payoff $r_{s,p}$ was defined as the expected value of the gamble selected by strategy $s$ on choice problem $p$, averaged across all 100 simulations. As in previous applications of the resource-rational strategy selection model [32, 39, 48], we quantified the expected cost of each strategy $c_{s,p}$ by counting the number of elementary information processes (e.g., reading or comparing; [49]) it required (see S5 Text for a detailed specification of the elementary information processes required by each strategy). While the strategies are essentially deterministic processes, ties were broken randomly; to take this randomness into account, we repeated the simulation of the strategies' choices 100 times and averaged $r_{s,p}$ and $c_{s,p}$ across all simulations.

The theory of resource-rational strategy selection assumes that a strategy's expected payoff $r_{s,p}$ and expected cost $c_{s,p}$ are approximated by internal predictive models based on the features of a choice problem ([32]; e.g., distribution of probabilities, similarity of attributes across options). These predictive models could be acquired, for instance, via reinforcement learning and be mentally represented as the weights of the features in the predictive model. This predictive model allows the decision maker to approximate a strategy's expected payoff and cost even without actually executing the strategy and also for new choice problems. In our modeling analysis, we make the simplifying assumption that the represented quantities are approximated by a given strategy's actual cost and actual payoff, but we do not explicitly model this approximation process.

The decision maker makes a choice with the selected strategy with a trembling-hand error $\epsilon$. With probability $1 - \epsilon$, the decision maker implements the choice predicted by the selected strategy; otherwise, the decision maker erroneously chooses the other option. The probability

of choosing option A over option B is therefore

$$P(A) = P(A|\delta, S) \cdot (1 - \epsilon) + P(B|\delta, S) \cdot \epsilon. \tag{2}$$

The inclusion of the trembling-hand error parameter was not preregistered. This modification of the model considerably improved model fit, especially with respect to reproducing the overall level of decision quality in the empirical data. As a consequence of this modification, we also adapted the model-estimation procedure slightly. The modifications to the model did not affect the conclusions to our preregistered hypotheses (see S6 Text for the results of the preregistered methodology).

To obtain the best-fitting parameter values for every participant, we performed a grid search across all possible strategy sets $S$ (ranging from a single strategy to up to seven strategies), values of $\delta$ from 0 to 20 (in steps of 0.1) and values of $\epsilon$ from 0.01 to 0.5 (in steps of 0.01). For each parameter combination, we computed the log-likelihood by simulating the model and deriving the probabilities of the observed choices according to Eq 2. To account for randomness in the model predictions due to random tie-breaking during both strategy selection and choice, we averaged the log-likelihood across 100 repeated model simulations. For each participant, we selected the parameter combination (including the composition of the strategy toolbox) that showed the maximum log-likelihood. If several parameter combinations showed the same maximum log-likelihood, we prioritized combinations with smaller sets of $S$ (i.e., smaller toolboxes) and then chose randomly between remaining parameter combinations. We report a parameter recovery analysis in S7 Text.

### Cumulative prospect theory

For the purpose of comparing the resource-rational strategy selection model with an established computational model of risky choice, we fitted cumulative prospect theory to the choice data of each individual participant. In cumulative prospect theory, the subjective value of a risky gamble is computed as the average of the gamble's nonlinearly transformed outcomes, weighted by a rank-dependent transformation of the outcome's probability. The value function for transforming the outcomes is characterized by two parameters, one representing the sensitivity to differences in outcomes ($\alpha$) and one representing differential weighting of gains and losses ($\lambda$). The probability-weighting function for transforming probabilities is characterized by two parameters representing the sensitivity to differences in probabilities, separately for gains and losses ($\gamma^+, \gamma^-$). The choice between two risky gambles is modeled with a softmax function, governed by a choice-sensitivity parameter ($\theta$). For a formal model description, we refer to [7].

We obtained maximum-likelihood estimates of the six model parameters for each participant by first performing a grid search across the complete parameter space (15 equally sized steps between the parameter boundaries $[0, 2]$). We then used the 30 best-fitting parameter combinations as starting points for an L-BFGS-B optimization algorithm to find the parameter combination with the maximum log-likelihood.

### Data analysis

We used the BayesFactor package [60] to compute Bayes factors and a prior concentration parameter of $a = 1$ for the Bayesian contingency table test. For Bayesian $t$-tests, we used Jeffreys-Zellner-Siow (JZS) priors with a scaling parameter of $r = \sqrt{2}/2$. For Bayesian correlation tests, we used stretched beta priors with a scaling parameter $\kappa = 1$. In the S1 Fig, we report Bayes factor robustness checks, varying the scaling parameter across wide ranges. Hierarchical regression models were computed with the brms package in R [61] using default priors.

## Supporting information

**S1 Text. Risk profiles of the strategies.**
(PDF)

**S2 Text. Are age differences in strategy selection mediated by age differences in affect?.**
(PDF)

**S3 Text. Application of the resource-rational strategy selection model to another dataset.**
(PDF)

**S4 Text. Comparison of the strategies' decision profiles.**
(PDF)

**S5 Text. Specification of strategy cost.**
(PDF)

**S6 Text. Preregistered version of the resource-rational strategy selection model.**
(PDF)

**S7 Text. Parameter recovery analysis.**
(PDF)

**S1 Fig. Bayes Factor robustness checks for the five hypotheses.**
(TIFF)

## Acknowledgments

We are grateful to Susannah Goss for editorial assistance.

## Author Contributions

**Conceptualization:** Florian Bolenz, Thorsten Pachur.

**Data curation:** Florian Bolenz.

**Formal analysis:** Florian Bolenz.

**Funding acquisition:** Thorsten Pachur.

**Investigation:** Florian Bolenz.

**Methodology:** Florian Bolenz.

**Project administration:** Florian Bolenz.

**Resources:** Thorsten Pachur.

**Software:** Florian Bolenz.

**Validation:** Florian Bolenz.

**Visualization:** Florian Bolenz.

**Writing – original draft:** Florian Bolenz.

**Writing – review & editing:** Florian Bolenz, Thorsten Pachur.

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
