## [Decision Letter · Decision Letter 0]

16 Jan 2024

Dear Dr Bolenz

Thank you very much for submitting your manuscript "Older Adults Select Different But Not Simpler Strategies Than Younger Adults in Risky Choice" for consideration at PLOS Computational Biology.

As with all papers reviewed by the journal, your manuscript was reviewed by members of the editorial board and by several independent reviewers. In light of the reviews (below this email), we would like to invite the resubmission of a significantly-revised version that takes into account the reviewers' comments.

As you can see the Reviewers expressed some serious concerns about the modelling results. In particular Reviewer 2 raises some important concerns about model validation that, if not addressed properly, will prevent the article being accepted. It is important to 1) show clear signatures of the models correctly fitting (or not) the data (see https://doi.org/10.1016/j.tics.2017.03.011) and 2) fit the model to all datasets. Of course, all the other issues raised by Reviewer 1 and Reviewer 2 have to be taken into account. Finally, please consider this https://psyarxiv.com/kyfus/download?format=pdf when preparing you rebuttal letter.

We cannot make any decision about publication until we have seen the revised manuscript and your response to the reviewers' comments. Your revised manuscript is also likely to be sent to reviewers for further evaluation.

Sincerely,

Stefano Palminteri

Academic Editor

PLOS Computational Biology

Daniele Marinazzo

Section Editor

PLOS Computational Biology

As you can see the Reviewers expressed some serious concerns about the modelling results. In particular Reviewer 2 raises some important concerns about model validation that, if not addressed properly, will prevent the article being accepted. It is important to 1) show clear signatures of the models correctly fitting (or not) the data (see https://doi.org/10.1016/j.tics.2017.03.011) and 2) fit the model to all datasets. Of course, all the other issues raised by Reviewer 1 and Reviewer 2 have to be taken into account. Finally, please consider this https://psyarxiv.com/kyfus/download?format=pdf when preparing you rebuttal letter.

Reviewer's Responses to Questions

**Comments to the Authors:**

Reviewer #1: Review of Older Adults Select Different But Not Simpler Strategies Than Younger Adults in Risky Choice by Bolenz et al.

Summary

In this study, the authors investigated strategies used by young and older adults in risky decision making. To that end, they re-analysed previously published data from an experiment where young and older participants made risky decisions in various choice problems. The novelty here is to explain subjects behaviour and strategy under the scope of the resource-rational strategy selection framework. Authors fitted the data with a model selecting, for each problem, one among different strategies (e.g. minimax) through a cost / expected benefits tradeoff. They show that older and younger adults tended to use qualitatively different set of strategies (i.e. with different risk propensities), while the set size, implementation errors, strategy complexity and sensitivity to cognitive costs were found to be similar in both groups.

The question investigated in the present study — What psychological mechanisms, cognitive or motivational, underly age difference in risky decision making? — is interesting and various analyses as well as a novel computational model have been introduced to answer it. The paper is very clearly written, with a compelling introduction and discussion. However, while the analyses provide some elements of an answer to the question, they are based on assumptions that appear unrealistic. Furthermore, some key control analyses are absent from the current version of the study, analyses that would ensure that the model is appropriate for describing participants’ strategies in the experiment. I introduce below a series of major and minor comments.

Major Comments

#1 Strategy overlap

From the current version of the paper, it is unclear what decisions are made overall by each of the strategies considered. Particularly, no detail is given about overlapping decision profiles between strategies. In other words, do the different strategies have clearly distinct decision patterns or is there a substantial overlap between them? Are some strategies indistinguishable in terms of decisions made in the set of problems? This is of prime importance considering that the model is fitted on the only binary choice made by subjects and that for each problem many strategies make the same decision. Could the authors provide details about the decision profile (across problems) of the strategies?

#2 “Strategy recovery”

This comment is related to the previous one and concerns the distinguishability of the strategies considered, in term of decisions produced. In the supplementary materials is presented a parameter recovery analysis which show that model parameters can be fairly well recovered. What is not clear is wether strategies themselves can be recovered. If I understand well, for a given strategy set size k, a same delta value implies that the exact same strategies will be selected by the model for all problems. However, slightly different delta value could imply different strategy selections. Could authors give details about the strategy profiles obtained with the generating parameters and with the recovered ones? Can strategies be fairly well recovered?

#3 Model conceptually hard to support

The key assumption of the resource-rational strategy selection model is that decision makers select a strategy based on a tradeoff of cognitive costs and expected benefits for each problem faced. In other words, they must compute the cost and expected value of each strategy in their “toolbox” before selecting the best one considering their sensitivity to cognitive cost. With this in mind, it seems peculiar to assume an account for cognitive cost in the selection, as it appears to have already been paid when computing the costs and benefits of each strategy. Could the authors give more details about this aspect of the decision and the realism of this assumption?

#4 Quality of fit

The model does not seem to be able to capture some parts of the data, particularly the choice quality of younger adults in losses and risk aversion of younger adults in gains. Could the authors explain why this is the case? What mechanisms or constraints prevent the model from accounting for these behaviours?

Minor Comments

#1 Strategy complexity check

In the discussion, authors emphasise the complexity of quantifying cognitive costs of the different strategies. Considering the method used here, is it possible to see in the data a relation between cognitive costs and strategy execution? Are most costly strategies more prone to execution error?

Reviewer #2: Bolenz & Pachur performed a re-analysis of a previously published dataset of younger and older adults who performed risky decision-making tasks (Pachur, Mata, Hertwig 2017). They applied the resource-rational computational model to investigate age-related differences in risky choices (risky vs. safe options). They observe that younger and older adults use different decision strategies (OA relied more on strategies focused on expected outcomes, whereas YA relied more on strategies focused on both probability and outcomes). They conclude that these age group differences reflect motivational differences between age groups.

Overall, although the authors present an interesting approach to address an important question regarding age differences in risky decision-making, it is not clear that the evidence presented supports the author’s claims. My main reservations are that 1) the model does not appear to fit the behavior, which limits the subsequent interpretations about parameter estimates associated with various cognitive strategies, 2) the age-invariance effects of the strategies typically associated with age are inconclusive (they are difficult in interpret given the clear age-related differences in behavior), 3) this model is only fit to one dataset, which limits the generalization of these claims. My comments are detailed below.

Major comments

- A primary concern is that the resource-rational strategy selection model used fit to the data does not provide strong evidence to support the author's claims. The model simulations mainly predicted age-invariant results but were not well-fit to the age-related differences in decision quality and risk aversion (e.g., in Figure 1). Additionally, given that the model revealed no significant age differences in key model parameters (e.g., strategy cost, toolbox size, cost-weighting parameters, error), the model results are rather inconclusive and difficult to interpret in terms of how they relate to cognitive (or motivational) differences between age groups.

- The model was only fit to one dataset, which limits the generalization of these findings. It would be helpful to replicate these modeling results in a different dataset with a similar task, which could provide more compelling evidence for model validation. It would also be helpful to compare the RR model to other extant models, which would demonstrate this model performs better than alternate models of risky DM.

- The authors argue that age-related differences are motivational in nature, but there is no evidence to support this claim. Given that the RR model dissects cognitive strategies, and that there is no explicit link to motivational factors (e.g., self-report, model parameters that reflect affective or motivational measures), these claims are not substantiated. Moreover, it is not clear whether OA considered outcomes (and not probability) due to capacity limitations (e.g., OA may only be able to process one dimension at a time vs. YA and can integrate both dimensions of outcome and probability more easily) or even attentional differences (e.g., OA prioritize outcome information vs. YA prioritize both). Overall, greater work is needed to demonstrate why these age-related differences relate to psychological factors involved in risky decision-making, thus limiting the impact of the current results.

**Have the authors made all data and (if applicable) computational code underlying the findings in their manuscript fully available?**

Reviewer #1: Yes

Reviewer #2: Yes

PLOS authors have the option to publish the peer review history of their article (what does this mean?). If published, this will include your full peer review and any attached files.

Reviewer #1: No

Reviewer #2: No
---

## [Decision Letter · Decision Letter 1]

28 May 2024

Dear Dr Bolenz

We are pleased to inform you that your manuscript 'Older Adults Select Different But Not Simpler Strategies Than Younger Adults in Risky Choice' has been provisionally accepted for publication in PLOS Computational Biology.

Before your manuscript can be formally accepted you will need to complete some formatting changes, which you will receive in a follow up email. A member of our team will be in touch with a set of requests. Please in submitting the final version consider addressing the following issue 

"It could be helpful to discuss how to improve how the RRSS model could be improved to better explain choice behavior, perhaps in a follow-up study."

Best regards,

Stefano Palminteri

Academic Editor

PLOS Computational Biology

Daniele Marinazzo

Section Editor

PLOS Computational Biology

Please in submitting the final version consider addressing in the discussion this point

"It could be helpful to discuss how to improve how the RRSS model could be improved to better explain choice behavior, perhaps in a follow-up study."

Reviewer's Responses to Questions

**Comments to the Authors:**

Reviewer #1: Review of Older Adults Select Different But Not Simpler Strategies Than Younger Adults in Risky Choice by Bolenz et al.

I thank the authors for their detailed answers to my concerns. They provided a pertinent set of analyses and details about the methods that substantially improves the clarity of the study.

Reviewer #2: The authors have sufficiently addressed all of the critiques, and their inclusion of additional analyses and model validation / posterior predictive checks have strengthened the paper (and support for the RRSS model for explaining cognitive mechanisms on age-related differences in risky choice). Overall, the paper addresses an interesting question regarding putative cognitive mechanisms underlying age-related differences in risky choice.

The only remaining outstanding question I have is the following: If the RRSS model is "better," compared to expected utility models, why is that the Cumulative Prospect Theory model better fit to the data? Is this due to overfitting of the model, or lack of regularization from the RRSS model (which presumably would worsen the log likelihood given the number of parameters for the more complex strategies). It could be helpful to discuss how to improve how the RRSS model could be improved to better explain choice behavior, perhaps in a follow-up study.

**Have the authors made all data and (if applicable) computational code underlying the findings in their manuscript fully available?**

Reviewer #1: Yes

Reviewer #2: Yes

PLOS authors have the option to publish the peer review history of their article (what does this mean?). If published, this will include your full peer review and any attached files.

Reviewer #1: No

Reviewer #2: No

---

## [Editor Report · Acceptance letter]

5 Jun 2024

PCOMPBIOL-D-23-01819R1 

Older Adults Select Different But Not Simpler Strategies Than Younger Adults in Risky Choice

Dear Dr Bolenz,

I am pleased to inform you that your manuscript has been formally accepted for publication in PLOS Computational Biology. Your manuscript is now with our production department and you will be notified of the publication date in due course.

With kind regards,

Zsofia Freund
